# Effects of Vertical Lifting Distance on Upper-Body Muscle Fatigue

**DOI:** 10.3390/ijerph18105468

**Published:** 2021-05-20

**Authors:** Nianli Fang, Chang Zhang, Jian Lv

**Affiliations:** 1Key Laboratory of Advanced Manufacturing Technology of the Ministry of Education, Guizhou University, Guiyang 550025, China; fnlalala@gmail.com; 2School of Mechanical Engineering, Southeast University, Nanjing 211189, China

**Keywords:** manual material handling, repetitive bending, sEMG, muscle fatigue, vertical lifting distance

## Abstract

Manual material handling (MMH) is commonly demanded in the manufacturing industry. Occupational muscle fatigue of the arm, shoulder, and back, which arise from MMH tasks, can cause work absences and low efficiency. The available literature presents the lack of the fatigue comparison between targeted muscles, on the same part or on different parts. The main aim of the present study was to evaluate and compare the fatigue of upper-body muscles during repetitive bending tasks, an experiment involving 12 male subjects has been conducted to simulate material handling during furniture board drilling. The vertical lifting distance was chosen to be the single independent variable, and the three levels were 0, 250, and 500 mm. Surface electromyography (sEMG) was used to measure the muscle fatigue of the biceps brachii, upper trapezius, and multifidus, while the sEMG parameters, including the normalized electromyographic amplitude (Normalized EA) and mean power frequency (MPF), of the target muscles were analyzed. The experimental results reveal that during the manual handling tasks, the biceps brachii was the most relaxed muscle, contributing the least muscle tension, while the multifidus was the most easily fatigued muscle. Furthermore, the EMG MPF fatigue threshold (MPFFT) of multifidus muscle tension was tested to estimate its maximum workload in the long-term muscle contraction. In conclusion, bending angle should be maintained to a small range or bending should even be avoided during material-handling tasks.

## 1. Introduction

In the traditional manufacturing industry, manual material handling (MMH) remains a common task, and bending motions are usually involved in MMH tasks. During repetitive bend-handling, the muscles on the arm, shoulder, and back frequently contract and easily fatigue, which even trigger work-related musculoskeletal disorders (WMSDs) for a long-term job [1,2]. Related research in China in the past decade shows that no fewer than 39% of manual operators in various industries such as mining, garment processing, and sonographing, have suffered from WMSDs, especially low back pain (LBP) and neck/shoulder pain (NSP) [3,4,5]. It is a remarkable fact that MMH tasks, as well as bend-handling motion, are generally involved in the above industries. What is worse is that most workers have no ergonomics learning experience.

Myoelectric manifestation of muscle fatigue, which signified the relationship between the work lord and the muscle fatigability, has been widely applied as an effective metric among studies toward work-related fatigue [6,7,8,9]. In this study, workload, which is indicated as the real-time tension of the muscle to sustain the external load [10,11], is regarded as an important factor in MMH task and a major cause of muscle fatigue [12]. Sustained muscle contraction, excessive muscle tension, and maintaining extreme working postures are all considered as the situations when local muscles suffer from abnormal external load. Therefore, the workload has been considered as muscle fatigue related risk factors during manual handling [13,14]. Specifically, as for the repetitive operation with high-frequency and low-load (for each time) during MMH tasks, the large vertical-distanced lifting is likely to cause myoelectric manifestation of muscle fatigue, even leading to WMSDs on the upper limb and shoulder [15,16,17]. Corresponding to workload, muscle fatigability is understood as the muscle endurance capacity to resist fatigue [18,19] or the ability to withstand the workload. The varying posture and contraction torque notably affect the muscle fatigability of the trapezius and multifidus during manual handling [20,21]. Otherwise, rotation of operating and resting in the process of the short-term MMH tasks, whose frequency varies corresponding to the workload, could reduce/increase muscle fatigue compared to performing a coherent heavier-load-leveled/lighter-load-leveled task without resting [22]. Moreover, during the sEMG detection, the impact of postural variation, compared to the task rotation, on the conventional manifestation of shoulder muscle fatigue is obviously easier to detect [23].

In this study, the muscle fatigability is analyzed by how the targeted muscle contributes its capacity, which is associated with how the workload of each muscle is. The workload factors mentioned above, divided in detail, include the duration, frequency, and torque of muscle contraction, as well as handling posture; all are associated with the transporting distance of the MMH task. Therefore, the vertical lifting distance, as the actual controllable variable during manufacturing, has been chosen as the single test factor in our experiment. At the same time, the handling load and frequency in our experimental tasks are set to a constant value.

Numerous studies toward work-related fatigue have targeted a single muscle or several muscles on a certain part [24,25], which lacks the fatigue comparison between targeted muscles, on the same part or on different parts. Therefore, this study has focused on the fatigue comparison between the targeted muscles, which are located on the upper-arm, shoulder, waist, respectively. All targeted muscles, including biceps brachii, upper trapezius, and multifidus, have proved to be sensitive to the fatigue-risk factors [15,21,22,26]. The fatigable muscle is capable of being discerned through comparing the myoelectric manifestation of muscle fatigue, while its capacity could be evaluated through the electromyographic mean-power-frequency fatigue threshold test (MPFFT). The MPFFT test for isokinetic muscle motions is an adaptation of the original EMG fatigue threshold (EMGFT) test [27,28], which is based on the linear decline of EMG spectral parameters during muscle fatiguing contraction. This linear decline trend is more significant than the linear increase trend of the amplitude parameters in the handling task of low-level workload [29,30].

In this study, a laboratory simulation of MMH tasks during the furniture manufacturing, restoring the work pace and cycle time at the investigated workshop, have been carried out. Among the simulation, a single-factor test was designed to examine the surface electromyography (sEMG) of three targeted muscles during the MMH tasks. Generally, the main purposes of the present study were: (1) to analyze the effect of the varying vertical distance on the tension and fatigability of each targeted muscle and to examine the difference in tension and fatigability of the targeted muscles in the same level of handling task; (2) to test the EMG MPF fatigue threshold for the most fatigable muscle of the targeted ones, and evaluate the capability of that muscle based on the fatigue threshold; (3) to reveal the fatigue development of the targeted muscles.

## 2. Materials and Methods

### 2.1. Subjects

The experiment simulated the board-handling task during the drilling process of furniture manufacturing (Figure 1), which was performed in the laboratory. We recruited subjects from local universities and screened out those who met the following requirements: (1) the anthropometric characteristics, such as age, height, weight, etc., is closed to the on-sited workers; (2) right-handed, since all the on-site workers are right-handed, and the myoelectric manifestation of the right-sided muscles were found to be more active [15,31]; (3) the targeted muscles: the biceps brachii, the upper trapezius, the multifidus, their strength, measured by the maximal voluntary contraction (MVC) test, was close to these workers; (4) without musculoskeletal disorders or injuries in the preceding six months and strenuous exercise within 24 h before the experiment; (5) except the training before the experiment, all of the subjects were non-experienced on the MMH tasks. At last, twelve eligible male subjects were selected to participate in our experiment (age: 23.7 ± 1.8 years; height: 175.0 ± 4.2 cm; weight: 67.5 ± 7.4 kg (mean ± SD)). All participants gave informed consent, which means that they knew about the experiment content and understood the risks involved.

This study was conducted according to the relevant regulations of the Ministry of Health of China “Methods of Ethical Inspection of Biomedical Research Involving People (trial)” and the Declaration of Helsinki on biological human trials. Meanwhile, it was approved by the Institutional Research Ethics Committee of Guizhou University (HMEE-GZU-2021-T002).

### 2.2. Preparation of Measurements

According to previous studies on the low-load repetitive operations in various force–velocity relationships [32,33,34], the biceps brachii, upper trapezius, and multifidus, which are directly involved in the operations and considered vulnerable to fatigue, were chosen to be the targeted muscles. There are several controllable variables as follows. Temperature: constant 27 °C; Work time: 8:00–12:00 a.m. or 14:00–18:00 p.m.; Dominant hand side: right.

The activity of targeted muscles, all on right side (the dominant hand side), was detected by sEMG (ErgoLAB, KING FAR, Beijing, China). Correspondingly, the sEMG data of muscles on the right side were analyzed after completing the detection. Before the electrodes were placed, the hair of related regions was shaved and the skin overlaying the targeted muscle was wiped with 75% medical alcohol. After that preparation, the skin impedance was reduced to 10 kΩ, which improved the signal-to-noise ratio (SNR) [35]. Then, the bipolar electrodes (SOLAR, China) were pasted to each muscle belly. Specifically, at 1/3 from the fossa cubit on the line between the medial acromion and the fossa cubit for the biceps brachii, at 50% on the line from the acromion to the spine on 7th cervical vertebra for the upper trapezius, and 2–2.5 cm right to the middle line of 5th lumbar vertebra spinous process for the multifidus. The reference electrode was placed on the radial condyle for the biceps brachii, the C7 spinous process for the upper trapezius, and the L5 spinous process for the multifidus [36,37]. Through the differential amplification and A/D (Analogue voltage to digital signal) conversion, the EMG signal was sampled at 1000 Hz with a bandpass filter at 10–500 Hz.

The MVC of the biceps brachii, upper trapezius, and multifidus was tested before all handling tasks (Figure 2). In the MVC test, the tested muscle intermittently contracted 3 rounds; a 3-second contraction and a 2-second interval were included in each round. The maximum among these three contractions was applied to normalize the EMG amplitude (EA) values in the sEMG test. After all the MVC tests were performed, the subject rested for 5–7 min until the potential of each muscle was completely restored [35]. Similarly, by the MVC tests and the comparison with the very first time, we were able to ensure that the subjects’ targeted muscles had been completely rested after the 2-or more-day break between two tested tasks.

### 2.3. Task and Measurements

When the task began, the subject brought the board (weight: 5.0 kg; size: 400 × 440 mm^2^) from the board rack to the workbench. As soon as the board was put on the workbench, the assistant transferred the board back to the rack (Figure 3). This pick-and-place process was continuously repeated 150 times in one handling cycle, and each pick-and-place (back and forth between the subject and the assistant) took 4 s (Figure 4). A metronome was used to guarantee a pick-and-place pace of 15 bpm. 

During the handling task, the subject and the assistant stand face to face with a distance of 600 mm as shown in Figure 3. The workbench was maintained at a height of 985 mm, while the board rack height was set at 985 mm in task VD0, 735 mm in task VD250, and 485 mm in task VD500, respectively. Hence, three levels of vertical distance were tested in the handling tasks: 0 mm, 250 mm, and 500 mm, which corresponded to three tasks above, respectively. Moreover, the distance between the workbench and the board rack, from one’s center point to the other’s, was 800 mm. In the process of the tested tasks, the ergonomic angle meter was applied to measure the subject’s bending angle. This instrument has been used like a protractor, one end was parallel to the thigh femur and one end was parallel to the back while measuring the bending angle.

### 2.4. Statistical Analysis

The sEMG data of muscles on the right side, which were considered to be the dominant ones among the MMH tasks, were analyzed. For each muscle, the average electromyography amplitude (EA) of the first three pick-and-place motions in each level handling task was recorded. Furthermore, the normalized average EA value represents the muscle tension of each load level, which is named MVE_0_%.
(1)$${\mathrm{MVE}}_{0}\%=\frac{{\mathrm{EA}}_{\mathrm{ACT}}}{{\mathrm{EA}}_{\mathrm{MVC}}}\cdot 100\%$$

For each tested task, the entire EMG signal was equally divided into 10 sections, while each of them contained 15 pick-and-place motions. The mean power frequency (MPF) of the EMG signal that corresponded to each pick-and-place motion was calculated using a fast Fourier transform (FFT) with a sliding window of 1000 samples and a 500-sample overlap between consecutive windows. Then, the average MPF of each section was obtained.
(2)$$\mathrm{MPF}=\int_{0}^{\infty }fPSD\left(f\right)df/\int_{0}^{\infty }PSD\left(f\right)df$$

With a linear fitting of 10 MPF values of each muscle in every task, the function of time was determined with slope coefficient and MPF-axis intercept (MPFs and MPF_0_, respectively) (Figure 5). MPFs is the muscle fatigue rate, and MPF_0_ is the initial MPF value in the task [38,39] (Figure 5). The muscle fatigability was examined by analyzing these two parameters.

Through one-way variance analysis (ANOVA) with repeated measures test, the effect of the vertical lifting distance on the muscle tension and muscle fatigue was studied. Simultaneously, the difference in muscle tension was analyzed by applying EMG MPFFT of the fatigable muscle. The MPF values in the first section of each task were linearly fitted, and the rate of the values decreasing with time (slope coefficient) was obtained [40]. The muscle tension levels in the tested tasks were plotted as a function of their corresponding slope coefficients for the MPF versus time relationships, while the EMG MPFFT was defined as the MPF intercept [41] (Figure 6).

## 3. Results

### 3.1. Analysis of Muscle Tension

The MVE_0_% values of the biceps brachii, upper trapezius, multifidus, from all subjects, in each tested task are shown in Figure 7.

The one-way ANOVA of the MVE_0_% values in three tested tasks shows that the vertical lifting distance has no significant effect on the biceps brachii muscle tension, nor on the upper trapezius muscle tension. But the vertical lifting distance significantly affects the multifidus muscle tension (*p* = 0.00 **). The pairwise inspection shows that the biceps brachii muscle tension remains at approximately 28~31% MVC in each task, whereas the upper trapezius muscle tension remains at 37~48% MVC. The multifidus muscle tension, which increases from 28% to 42% MVC with the increase of height, is significantly stronger in task VD500 than in tasks VD0 (*p* = 0.00 **) and VD250 (*p* = 0.00 **). There is no significant difference in the tension of each muscle between task VD0 and VD250.

The difference analysis of the _0%_ values of three targeted muscles indicates no significant difference in task VD0, as well as in task VD250. However, there is a significant difference between three muscles in task VD500 (*p* = 0.00 **). Specifically, the pairwise inspection reflects that the biceps brachii and the multifidus have similar muscle tension in tasks VD0 and VD250. Furthermore, the multifidus contributed significantly greater muscle tension than the biceps brachii in task VD500 (*p* = 0.00 **). In all three tasks, the upper trapezius generated relatively higher muscle tension than the biceps brachii; the significance was only found in task VD500 (*p* = 0.00 **). Similarly, its muscle tension was relatively higher than the multifidus; the significance was only found in task VD250 (*p* = 0.05 *).

### 3.2. Analysis of Muscle Fatigue

#### 3.2.1. Analysis of the Initial MPF Value

The MPF_0_ values of the biceps brachii, upper trapezius, and multifidus, from all subjects, in each tested task are shown in Figure 8.

The one-way ANOVA of the MPF_0_ values in the tested tasks shows that the vertical lifting distance has no significant effect on the initial MPF value of the biceps brachii and upper trapezius but significantly affects the initial MPF value of the multifidus (*p* = 0.01 **). The pairwise inspection shows that the MPF value of the biceps brachii remains at approximately 73–75 Hz in each task and that of the upper trapezius remains at 75–76 Hz. Besides, the initial MPF value of the multifidus, which increases from approximately 88 Hz to 109 Hz with increasing height, was significantly lower in task VD500 than in task VD0 (*p* = 0.00 **) and VD250 (*p* = 0.01 *). There was no significant difference in value in task VD0 and VD250.

The difference analysis of the MPF_0_ values of all the targeted muscles indicates significant difference in task VD0 (*p* = 0.00 **), as well as in task VD250 (*p* = 0.00 **) and in task VD500 (*p* = 0.02 *). In detail, the pairwise inspection reflects that the biceps brachii and the upper trapezius have similar initial MPF values in each tested task. However, the multifidus has a significantly higher initial MPF value than the biceps brachii (VD0: *p* = 0.00 **, VD250: *p* = 0.00 **, VD500: *p* = 0.01 *) and the upper trapezius (VD0: *p* = 0.00 **, VD250: *p* = 0.00 **, VD500: *p* = 0.02 *) in all 3 tested tasks.

#### 3.2.2. Analysis of Fatigue Rate

The MPF values of the biceps brachii, upper trapezius, and multifidus in each tested task, from all subjects, are shown in Figure 9.

The one-way ANOVA of the MPF values in the tested tasks shows that the vertical lifting distance has no significant effect on the fatigue rate of the biceps brachii, nor on the upper trapezius, but it significantly affects the fatigue rate of the multifidus muscle (*p* = 0.01 **). The pairwise inspection shows that the muscle fatigue rates of the biceps brachii and upper trapezius scatter in different ranges in each task. There is no significant difference between any two tasks, except that the upper trapezius has a significantly higher fatigue rate in task VD0 than in task VD500 (*p* = 0.03 *). For the multifidus, the fatigue rate of task VD500, approximately decreasing from −0.31 to −0.61 Hz/min with increasing height, is significantly lower than that of task VD0 (*p* = 0.01 **) and task VD250 (*p* = 0.04 *). There is no significant difference within its fatigue rates between task VD0 and VD250.

The difference analysis of the MPF values of the tested muscles indicates significant differences in task VD0 (*p* = 0.00 **), as well as in task VD250 (*p* = 0.00 **) and VD500 (*p* = 0.00 **). Specifically, the pairwise inspection reflects that the biceps brachii has a higher muscle fatigue rate than the multifidus (VD0: *p* = 0.00 **, VD250: *p* = 0.00 **, VD500: *p* = 0.00 *), and upper trapezius (VD0: *p* = 0.35, VD250: *p* = 0.04 *, VD500: *p* = 0.02 *), in each tested task, while the multifidus muscle always perform the lowest fatigue rate among three targeted muscles (for the multifidus and the upper trapezius, VD0: *p* = 0.00 **, VD250: *p* = 0.03 *, VD500: *p* = 0.00 *). Besides, for the biceps brachii and the upper trapezius, there is no significant difference between their muscle fatigue rates in task VD0.

#### 3.2.3. Calculation of EMG Fatigue Threshold

The multifidus muscle is chosen to test its EMG MPF fatigue threshold. This way, we are able to estimate its maximum workload in the long-term muscle contraction. The EMG MPFFT of each subject is shown in Table 1.

The EMG MPFFT of all subjects can be divided into two types:(1)The EMG MPFFT value is between the MVE_0_% values of tasks VD0 and VD250.(2)The EMG MPFFT value is less than the MVE_0_% value of task VD0.

## 4. Discussion

### 4.1. Comparison of Muscle Fatigue

Within the handling tasks, the varying vertical lifting distance has a non-significant effect on the muscle tension of the biceps brachii and on its initial MPF value and fatigue rate. With increasing vertical lifting distance, the muscle tension of biceps brachii maintains a low level, whereas the fatigue rate is mostly positive and always the highest among three tested muscles. Thus, the biceps brachii is hardly affected by the waist-bending motion and is non-susceptible to fatigue. In fact, during the MMH tasks, the biceps brachii has been demonstrated to be the most active one among the arm muscles because of its upper extremity position during lifting and handling [42]. The above situation can be explained as follows: how the upper-body was postured, particularly for the shoulder, during the manual handling task ensured that the biceps brachii muscle was contracting isotonically for most of the time during the handling tasks [43,44]. Therefore, its muscle torque remained in a stable range [45,46].

The myoelectric manifestation of the shoulder muscles, particularly the trapezius, expresses sensitively to the lifting and handling motion. With the increase in vertical lifting distance, the upper trapezius muscle tension remains at a medium level, which is also the highest among the target muscles. Actually, it has been proved that even in low-intensity work such as light-assembly work, an increase in the amplitude and a decrease in the frequency of the trapezius muscle EMG could be observed [32,47,48]. The fatigue rate of the upper trapezius muscle significantly decreases when the lifting height, the single variable, is maximal. Thus, during the handling tasks, the upper trapezius shares more workload than the biceps brachii. Correspondingly, due to the change of muscle torque, the upper trapezius becomes obviously fatigable, and more easily, while the operators bend down repeatedly.

High individual differences are observed for the fatigue rates of the biceps brachii and upper trapezius in each tested task, which might be explained by the difference in muscle capacity of the subjects [19].

For the multifidus involved in MMH tasks, with the increase in vertical lifting distance, its muscle tension increases in a certain range: from low to medium. Meanwhile, its initial MPF value decreases but remains the highest among three targeted muscles, and the fatigue rate significantly decreases. Therefore, the multifidus, which shares the largest workload and the largest torque among three targeted muscles, its muscle fatigue is significantly affected by the waist-bending motion during the handling tasks. Thus, within MMH tasks that are similar to our experimental ones, the multifidus is likely to be the most susceptible to fatigue, while the accumulated fatigue causes the muscle to hardly recover within the work time [49,50].

During MMH tasks, most operators are accustomed to bending over and lifting up/laying down boards. This direct bending-and-lifting motion is likely to cause upper-body muscle fatigue and is harmful to the spine and the back muscles. It is recommended to lift the load in a semi-squatting posture with the back straight, which is able to protect the spine and improve the torque and strength exertion of upper-body muscle [23,51,52,53,54]. Furthermore, the applying of the work chair, which provides support for the operator’s buttocks, would allow the operator to maintain a working posture between standing and leaning. This makes it possible to relieve the back muscle during the light-load handling task.

### 4.2. Discussion of the EMG Fatigue Threshold

The analysis of the EMG MPFFT of multifidus shows that the value is between the MVE_0_% values of task VD0 and task VD250 or less than the MVE_0_% value of task VD0, which suggests reducing or avoiding the vertical lifting distance during the long-time manual handling tasks, i.e., to minimize the bending motion in the pick-and-place task. By applying the ergonomic angle meter, it was found that the waist of the subject was approximately bending by 25° to 40° in test V250. This suggests that the waist bending angle of the operator should be less than that range above during long-time MMH task. Unfortunately, EMG parameters such as EA, MPF, as well as Borg’s perceived exertion and pain scales, cannot describe the fatigue development in numerous low-load or intermittent tasks [23,51,52,53,54]. Nevertheless, the EMG MPFFT fatigue threshold, which indicates the maximum workload on the muscle in the long-term contraction, can be considered as an effective reference for the load arrangement and the workload design.

It can be seen from Figure 10, the operator is able to avoid repetitively bending-down-to-pick and straightening-up-to-place, as the height of the boards rack is always maintained slightly lower than the elbow height. According to that, Yu and Zhang et al. designed a bend-free boards rack that combines infrared positioning and hydraulic lifting [47]. As shown in Figure 11, with the cooperation of infrared-emitting and-receiving tubes, signal processor, and various valves inside the hydraulic system, the upper shelf of the rack is able to maintain at the fixed height. Furthermore, one can remove the board neatly when the inside of the rack is full. After resetting the infrared signal processor, the motor-driven lifter rises back to the top.

## 5. Limitations and Future Work

There are a few limitations that should be considered while interpreting the results of this study. As for the comparison of muscle fatigue between different muscles, the muscles are located in different parts of the upper-body parts. The purpose of the targeted muscles selecting is to have a good understanding of the contribution of muscle capacity in each body part. Sequentially, the muscle fatigability of the targeted muscles is able to be compared. It should be recognized that the task performance is the only factor that has been considered, while the fiber composition and contraction type are not taken into account. Though the physical factors such as fiber composition greatly affect the myoelectric manifestation, the applying of normalized value of myoelectric parameter, which figures out the capacity contribution of each muscle, is able to avoid the influence of these factors up to a point.

In order to reflect the real working environment, the vertical lifting height is chosen to be a single variable in the MMH experiment. However, while modifying that variable as a comprehensive factor, several workload factors, including the holding duration, motion frequency, and torque of muscle construction, should be considered and well-designed, especially when the EMG MPFFT value of the muscle is excessively small. Specifically, handling loads of varying weights, heavier/lighter than 5.0 kg, with a reasonable bending angle would be experimented with during follow-up studies.

Previous research findings into the difference between the left and right side of the body have been inconsistent and contradictory [15,31,55]. A number of studies have found that the myoelectric manifestation of the right-sided muscles was more active, and its changing trend is better to analyze [15,31]. Prior research showed that no significant difference was observed between the left and right side in sEMG signals [7]. However, an initial pilot study found no significant difference between onset latencies for the right and left paraspinal muscles during sudden loading [55]. Therefore, sEMG recording was restricted to the right side of the body in this research. Further research should be done to investigate the difference between the right- and left-sided muscle fatigue in the future. Although the manifestation of muscle fatigue has been defined as a combination of the increase in EMG amplitude and decrease in frequency, it is expected that one will perceive an accurate fatigue development applying EMG parameters [39,56]. As an unconventional reference for muscle fatigability analyzing and workload controlling, the EMG MPFFT fatigue threshold would be more practical while combining with the joint analysis of EMG spectrum and amplitude (JASA). The JASA method is capable of understanding the real-time muscle fatigability, which would be more instructional for the task variation, as well as the short-term/long-term load distribution [57].

## 6. Conclusions

The objective of this paper was to explore the relationship between the fatigue of upper-body muscles and repetitive bending tasks as well as provide useful and specific guidelines for performing repetitive bending tasks.

The main findings are as follows: In the real-workplace-restored MMH tasks, the multifidus proved to be the most fatigable, while the biceps brachii was the most relaxed. Referring to the actual operating situation, the waist-bending and the shoulder motion during lifting and handling hardly affects the arm muscle contraction, which results in the best load capacity of the arm muscles. Meanwhile, vertical lifting distance significantly contributes to back muscle contraction, which results in the worst load capacity.

These facts lead us to suggest that lifting the load with small-angled bending, or even avoiding passive bending over, would effectively protect the upper-body muscles during MMH tasks. How to optimize the task variation, as well as the short-term/long-term load distribution, will be explored during follow-up studies.

## Figures and Tables

**Figure 1 ijerph-18-05468-f001:**
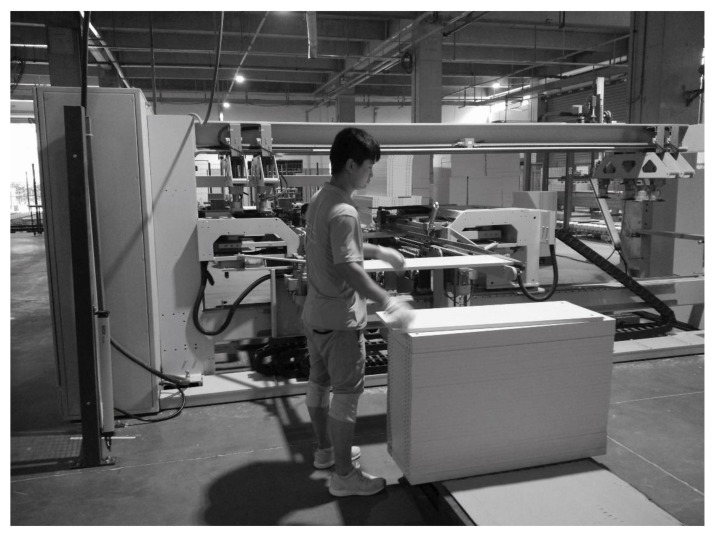
Handling panel boards to the inlet of six-row drilling machine (photo from a furniture manufacturing factory in China).

**Figure 2 ijerph-18-05468-f002:**
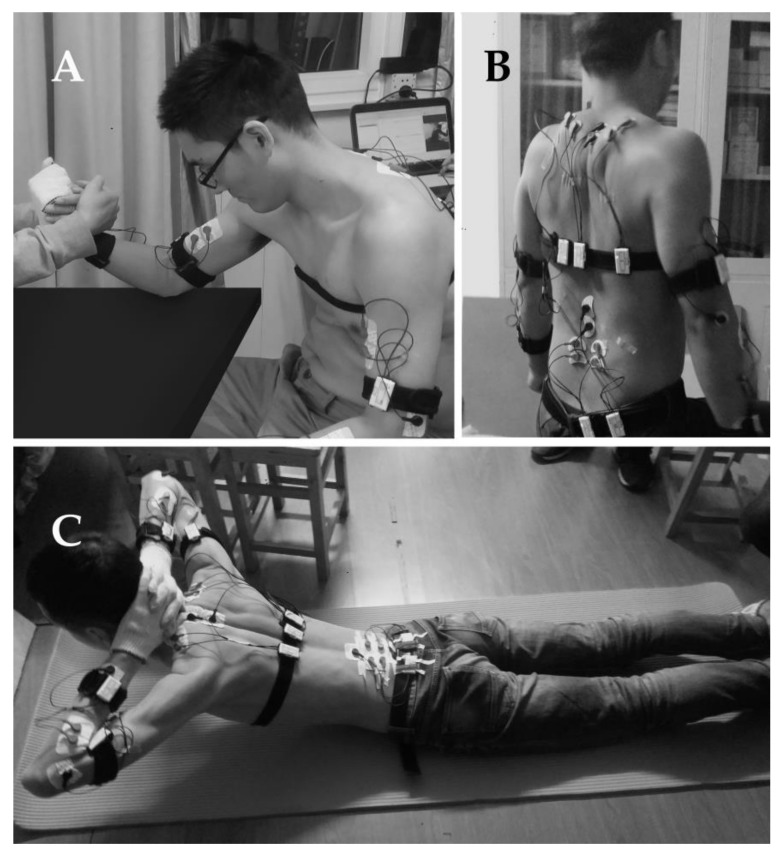
The MVC test for the muscles [35]: (**A**) MVC test for biceps brachii muscle. The subject endeavored to flex the arm while in a seated position and his forearm was held tightly. (**B**) MVC test for upper trapezius muscle. The subject lifted the shoulder utmost while his arms were fixated. (**C**) MVC test for multifidus muscle. The subject endeavored to raise the chest laying in a prone position and his ankles were held down.

**Figure 3 ijerph-18-05468-f003:**
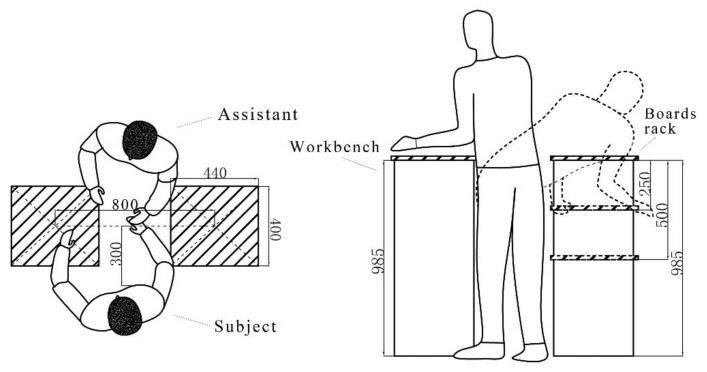
Illustration of the handling experiment layout and the test variable design: the subject and the assistant stand face to face with a distance of 600 mm, between them the workbench and the boards rack placed in a line. While the workbench is set at a fixed height, the sheets rack is set at three height levels. The subject moves the board from the rack to the workbench, while the assistant transports it in the opposite direction.

**Figure 4 ijerph-18-05468-f004:**
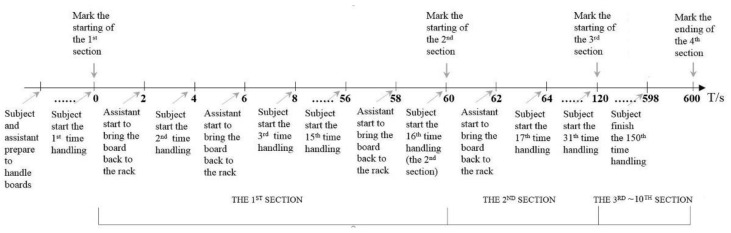
Illustration of the repetitive material handling process.

**Figure 5 ijerph-18-05468-f005:**
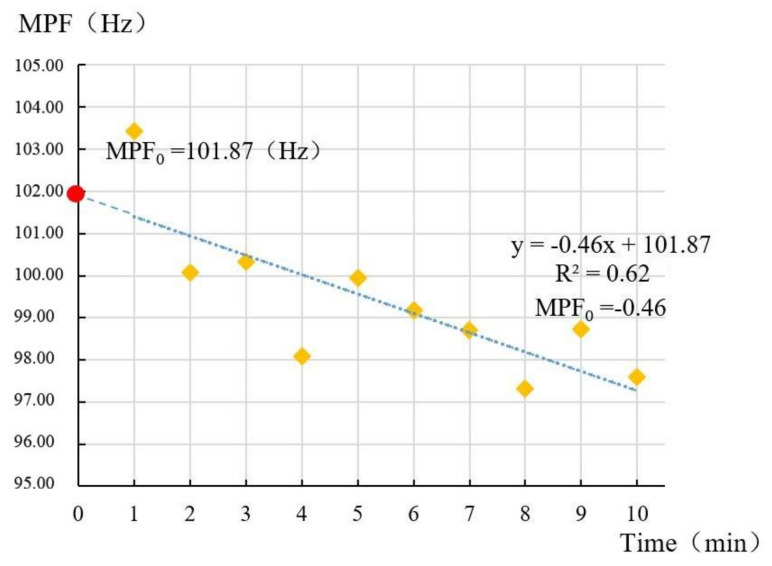
Linear regression of the MPF of the multifidus muscle in task VD250 (subject 1). The MPF data (10 sections) are plotted as a function of time (min) for each task, and the slope coefficient and MPF−intercept were determined.

**Figure 6 ijerph-18-05468-f006:**
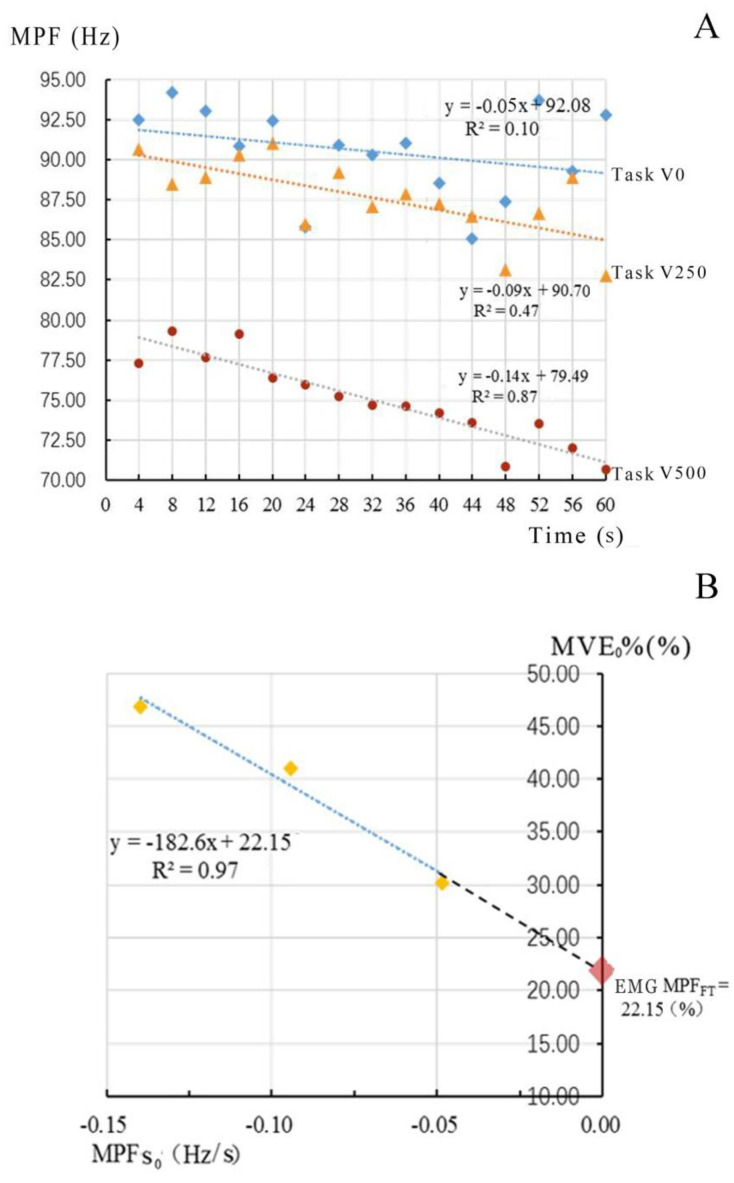
Process of measuring EMG MPFFT from data of the multifidus muscle of subject 8. (**A**) The MPF data (10 sections) were plotted as a function of time (s) for 3 muscle tensions (30.23%, 41.98%, 47.16%MVC in the figure). (**B**) The MPF slope coefficients obtained from (**A**) were plotted versus muscle tension for each test level and the EMG MPFFT was measured as the y−intercept value (22.15%).

**Figure 7 ijerph-18-05468-f007:**
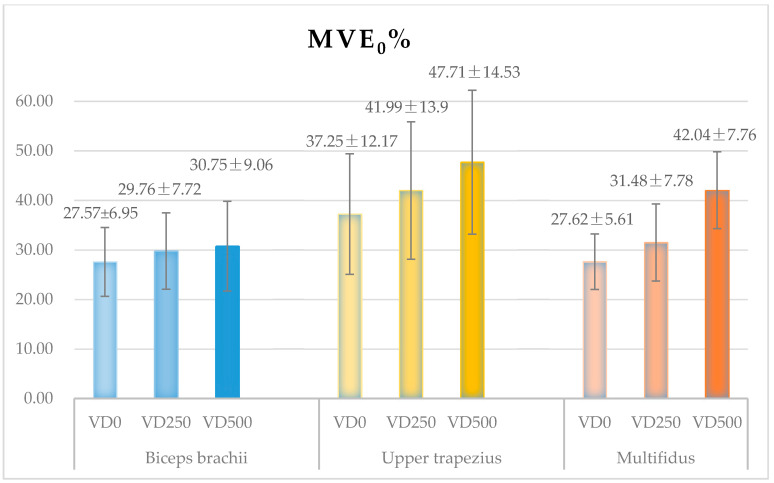
The tension values (MVE_0_%) of target muscles (biceps brachii, upper trapezius, multifidus), combined all the subjects, in three tested tasks.

**Figure 8 ijerph-18-05468-f008:**
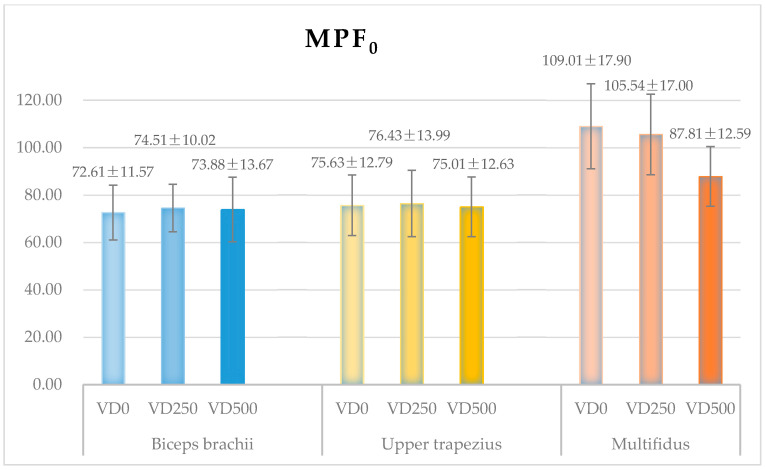
Initial MPF values (MPF_0_) of target muscles (biceps brachii, upper trapezius, multifidus), combined all the subjects, in three tested tasks.

**Figure 9 ijerph-18-05468-f009:**
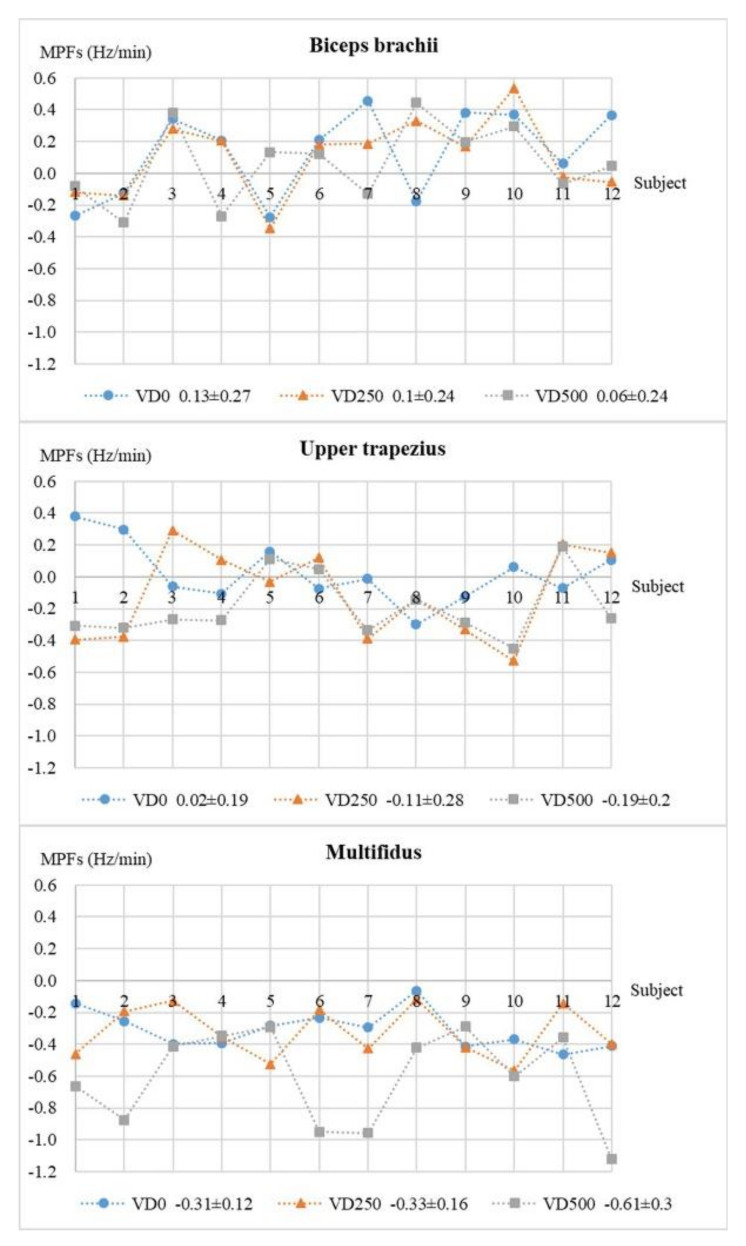
The fatigue rates (MPFs) of target muscles (biceps brachii, upper trapezius, multifidus), combined all the subjects, in three tested tasks.

**Figure 10 ijerph-18-05468-f010:**
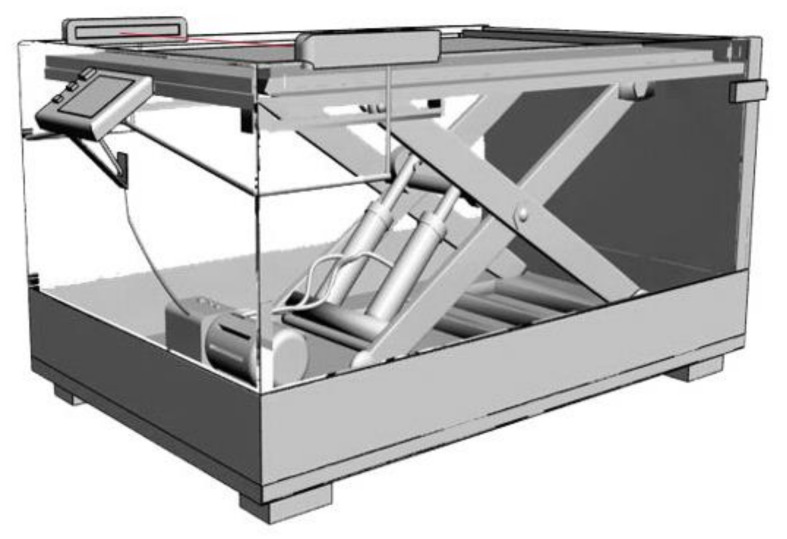
The bend-free boards rack, equipped with infrared positioning device and hydraulic actuator.

**Figure 11 ijerph-18-05468-f011:**
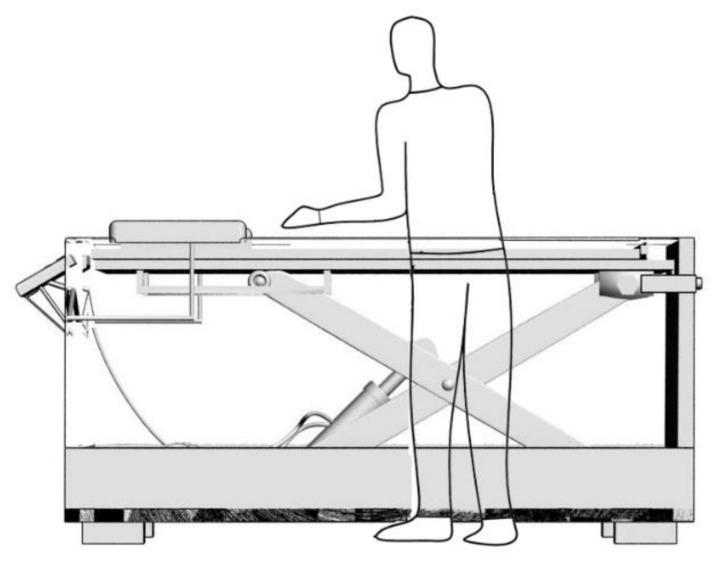
Application of the bend-free boards rack. The uppermost plane of the board pile maintains at the lower end of the elbow, which is set by 5th percentile (96.5 cm for Chinese male). The board is sized no more than 1220 × 610 mm^2^ and the number of 18 mm boards is no more than 50 pieces.

**Table 1 ijerph-18-05468-t001:** The EMG MPFFT and the original muscle tension of the multifidus muscle of each subject in each tested task.

Subject No.	EMG MPFFT (%)	MVE_0_% (%)		
		VD0	VD250	VD500
1	37.16	35.86	37.89	41.74
2	28.47	33.97	43.06	41.52
3	28.9	33	38.71	52.55
4	22.68	30.28	34.65	50.86
5	10.36	19.72	23.98	47.46
6	14.11	20.53	19.36	26.88
7	31.05	28.31	31.3	40.16
8	22.15	30.23	40.98	46.85
9	10.68	22.48	23.09	35.9
10	7.6	20.58	23.88	47.07
11	21.1	29.95	32.03	42.69
12	13.41	26.56	28.8	30.87
$\bar{x}$ ± σ	21.47 ± 9.22	27.62 ± 5.61	31.48 ± 7.78	42.04 ± 7.76

## Data Availability

Data available on request due to privacy and ethical restrictions. Main data is contained within the article.

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
