# Peer review of "Effects of Vertical Lifting Distance on Upper-Body Muscle Fatigue"

_ijerph, 2021, doi:10.3390/ijerph18105468_

Round 1
Reviewer 1 Report
I have no more comments
Author Response
This is the newly revised version of our manuscript. Thanks for all you suggestions!

Reviewer 2 Report
Dear authors,
thank you for taking the time to revise the manuscript and to answer all my queries. However, I cannot advise to accept the paper, since, for publication, a vote from an ethics committee is mandatory. I also do not agree, that the risk for the subjects was minimal. You write, that not all of them were physically active and you tried to fatigue the muscles of the back. Therefore, there is a (small) risk for back strain, lumbago or other complaints in the back.
The other reason, why I decline to accept is, that you did not standardize the lifting procedure. For example, you could have done a screening on how workers usually perform the task and then advise your subjects to do it the most common way. But like this, scientific soundness is not given. Moreover, you cannot just exclude the results from the left-hand bodyside, you would have to show them in the results and discuss them later. It would have been nice if the lifting task would have been performed as well on the left side.
Author Response
Dear Reviewer:
We would like to thank all the constructive comments, again. Your suggested revisions have been carefully considered, and the manuscript was revised accordingly.
The following are the replies to the comments.
Thank you for taking the time to revise the manuscript and to answer all my queries. However, I cannot advise to accept the paper, since, for publication, a vote from an ethics committee is mandatory.
Reply: L104-107—We have obtained approval from the Institutional Research Ethics Committee of Guizhou University (HMEE-GZU-2021-T002) in time. Meanwhile, the appendix file could confirm that the whole project, especially the experiments, was in accordance with the relevant regulations of the Ministry of Health of China 《Methods of Ethical Inspection of Biomedical Research Involving People (trial)》and the Declaration of Helsinki on biological human trials. We sincerely hope this declaration, and the relevant files could eliminate your doubt about the ethicality of this project.
I also do not agree, that the risk for the subjects was minimal. You write, that not all of them were physically active and you tried to fatigue the muscles of the back. Therefore, there is a (small) risk for back strain, lumbago or other complaints in the back.
Reply: The reason that we capable of guaranteeing the minimal risk is that we have carried out a pre-experiment before the formal experiment, which was approved by the ethical committee, as well. Through the pre-experiment, we determined the workloads within the formal experiment. Otherwise, we found that a short-term muscle ache on the shoulder and the back, which could be totally recovered within 48 h, is possible. Moreover, none subject in the pre-experiment has suffered from muscle strain, lumbago, or other complaints on any body part, since a 10-day gap, between the pre-and the formal one, allows us to observe the subjects and confirm the above results.
The other reason, why I decline to accept is, that you did not standardize the lifting procedure. For example, you could have done a screening on how workers usually perform the task and then advise your subjects to do it the most common way. But like this, scientific soundness is not given.
Reply: Apologies for that we omit describing the standardized design of the MMH tasks, while we do standardize its procedure. Firstly, we straightly set the height of the workbench to be the same as in the workshop we researched, while the position of the boards rack, the distance between the boards rack and the workbench is determined by the average angle of operators’ sagittal plane rotation (by applying the ergonomic angle meter) and the average distance operators carrying a board. Secondly, the subjects chosen are somewhat similar to the on-sited workers in terms of anthropometric characteristics: height, weight, muscle strength(by the MVC test), since that we can’t recruit the on-sited operators directly. Thirdly, we restore the on-sited operators’ typical habitual actions during the lifting-and-handling procedures, such as seldom bending the back (not the waist) and the knees, seldom moving the shoulder and the upper arm. Hope this description could express our MMH task standardization well.
Moreover, you cannot just exclude the results from the left-hand bodyside, you would have to show them in the results and discuss them later. It would have been nice if the lifting task would have been performed as well on the left side.
Reply: Previous research findings into the difference between the left and right body side have been inconsistent and contradictory[1-3]. In this paper, all right-handed 12 subjects are chosen to participate in the experiment since the myoelectric manifestation of the right-sided muscles was more active to be found [3, 2]. Moreover, we’ve tried to discuss the findings of the left body side and the right body side in the discussion. Further research should be done to investigate the difference between right-sided and left-sided muscle fatigue in the future.
The content we added is as follows
Line 91-93: Twelve right-handed male subjects are selected to participate in the experiment since the myoelectric manifestation of the right-sided muscles was more active to be found [3, 2]
Line 367-374: Previous researches findings into the difference between the left and right body side have been inconsistent and contradictory[1-3] A number of studies have found that the myoelectric manifestation of the right-sided muscles was more active to be found[2, 3]. Prior research showed that no significant difference was observed between the left and right side in sEMG signals[4]. However, an initial pilot study found no significant difference between onset latencies for the right and left paraspinal muscles during sudden loading [1]. Therefore, sEMG recording was restricted to the right side of the body in this research. Further research should be done to investigate the difference between the right- and left-sided muscle fatigue in the future.
Line 393-395: There is abundant room for further progress in investigating the relationship between the left side and right side in MMH task.
Reference:
[1]MAWSTON G, MCNAIR P, BOOCOCK M. The effects of prior warning and lifting-induced fatigue on trunk muscle and postural responses to sudden loading during manual handling [J]. Ergonomics,2007, 50(12): 2157-2170.
[2]RANAVOLO A, CHINI G, SILVETTI A, et al. Myoelectric manifestation of muscle fatigue in repetitive work detected by means of miniaturized sEMG sensors [J]. International Journal of Occupational Safety and Ergonomics,2018, 24(3): 464-474.
[3]THEUREL J, DESBROSSES K, ROUX T, et al. Physiological consequences of using an upper limb exoskeleton during manual handling tasks [J]. Applied ergonomics,2018, 67: 211-217.
[4]ANTWI-AFARI M, LI H, EDWARDS D, et al. Biomechanical analysis of risk factors for work-related musculoskeletal disorders during repetitive lifting task in construction workers [J]. Automation in Construction,2017, 83: 41-47.

Reviewer 3 Report
The authors have responded properly to most of comments made in previous review (R0).
It is only necessary to point out the lack of response to the following comments:
Introduction
- Objective 3 “And to explore methods to relieve the muscle fatigue during handling tasks based on its development” is not possible to meet, because the methodological design and results obtained do not allow for it. In my opinion, it should be removed.
Methodology
- The study design remains undefined. Reference is made to the recruitment system (random), but it is not explained how this randomization was carried out.
- Concerning Ethical Considerations, only obtaining the “informed consent” of the participants is mentioned. It should be included the approval by the corresponding Ethics Committee (with registration number). In case it is not necessary (whatever the reason), I think it´s necessary to indicate it in the text or in a separate response document.
Conclusions
- This paragraph “These facts lead us to suggest that lifting the load with a semi-squatting posture, or small-angled bending, would effectively protect the upper-body muscles during MMH. Moreover, the application of bend-free boards rack is able to place and remove the board in a more effortless way. How to optimize the task variation, as well as the short-term / long-term load distribution will be explored during following-up studies” should be exclusively part of the Discussion (in part it is already included), since it is not possible to draw directly from the study
Concerning the new revised version of the manuscript, I would add the following comments:
- Line 40-41: “work lord” ???:
- Paragraph lines 40-42. There seems to be a lack of bibliographic reference (s) that support this claim.
- Lines 56-58: I do not understand this sentence. Is it possible to re-write it to make it easier to understand?
- Lines 115-116: The placement of the electrodes on the biceps brachii remains to be explained.
- Line 126: “By” in capital letters.
- Figure 7 and 8. Do these graphs refer to the combined data of all study subjects? Actually this is not explained neither in the text nor in the figure caption / s.
In addition, it would be very visual (to facilitate understanding of the work), to indicate on the graph (eg with asterisks) the statistically significant differences between heights (in the same muscle) and between muscles (for the same height).
- Lines 262-264: Indicate at the end of line 263 the number of subjects (2 subjects) and at the end of line 264 (10 subjects).
- Line 299-300. Please check this sentence (syntactic construction). It's not easy to understand.
- Fig 10. Unnecessary. Furthermore, it is not cited in the text. Please remove it.
Although I think it is correct to discuss the existence of the “bend-free board-rack design”; however, I believe that such a detailed explanation of its performance is unnecessary (i.e., please simplify lines 330-335). Also, I would suggest to remove Figures 11 and 12, as they do not seem to add anything to the text of the manuscript.
Author Response
Dear Reviewer:
We would like to appreciate for all the constructive comments, again. Your suggested revisions have been carefully considered, and the manuscript has been revised accordingly.
The following is the replies to the comments. (The line numbers we’ve marked only correspond to the pdf. file.)
Introduction:
- Objective 3 “And to explore methods to relieve the muscle fatigue during handling tasks based on its development” is not possible to meet, because the methodological design and results obtained do not allow for it. In my opinion, it should be removed.
REPLY: L87-- After reconsideration, we decided to remove this portion according to your suggestion. And it turns out to be more logical in context.
Methodology:
- The study design remains undefined. Reference is made to the recruitment system (random), but it is not explained how this randomization was carried out.
REPLY: L91-99, P2-3-- We are terribly sorry that our vague expression caused your misunderstanding. In fact, we recruited subjects from local universities, and screened out those who met the following requirements: 1) the anthropometric characteristics, such as age, height, weight, etc., is closed to the on-sited workers; 2) right-handed, since all the on-sited workers are right-handed, and the myoelectric manifestation of the right-sided muscles were found to be more active; 3) the targeted muscles: the biceps brachii, the upper trapezius, the multifidus, their strength, measured by the maximal voluntary contraction (MVC) test, was close to these workers; 4) without musculoskeletal disorders or injuries in the preceding six months and strenuous exercise within 24 hours before the experiment; 5) except the training before the experiment, all of the subjects were non-experienced on the MMH tasks. So that this recruiment process is not random at all.
-Concerning Ethical Considerations, only obtaining the “informed consent” of the participants is mentioned. It should be included the approval by the corresponding Ethics Committee (with registration number). In case it is not necessary (whatever the reason), I think it´s necessary to indicate it in the text or in a separate response document.
REPLY: L103-106-- We have obtained the approval from the Institutional Research Ethics Committee of Guizhou University (HMEE-GZU-2021-T002) in time. Meanwhile, the appendix file could confirm that the whole project, especially the experiments, was in accordance with the relevant regulations of Ministry of Health of China 《Methods of Ethical Inspection of Biomedical Research Involving People (trial)》and the Declaration of Helsinki on biological human trials. We sincerely hope this declaration, and the relevant files, could eliminate your doubt about the ethicality of this project.
Discussion:
- This paragraph “These facts lead us to suggest that lifting the load with a semi-squatting posture, or small-angled bending, would effectively protect the upper-body muscles during MMH. Moreover, the application of bend-free boards rack is able to place and remove the board in a more effortless way. How to optimize the task variation, as well as the short-term / long-term load distribution will be explored during following-up studies” should be exclusively part of the Discussion (in part it is already included), since it is not possible to draw directly from the study
REPLY: L392-393-- We agreed with your suggestion and removed such exploratory conclusion, just kept “These facts lead us to suggest that lifting the load with small-angled bending, or even avoid passive bending over, would effectively protect the upper-body muscles during MMH tasks.”
Concerning the new revised version of the manuscript, I would add the following comments:
Line 40-41: “work lord” ???:
REPLY: All the “work load”s have been corrected to “workload”.
Paragraph lines 40-42. There seems to be a lack of bibliographic reference (s) that support this claim.
Reply: L41-43-- Some references have been added in this paragraph.
“Myoelectric manifestation of muscle fatigue, which signified the relationship between the work load and the muscle fatigability, has been widely applied as an effective metric among researches toward work-related fatigue[6-9]”.
[6] Äng, B., J. Linder, and K. Harms-Ringdahl, Neck strength and myoelectric fatigue in fighter and helicopter pilots with a history of neck pain. Aviation, space, and environmental medicine, 2005. 76(4): p. 375-380.
[7] Antwi-Afari, M., et al., Biomechanical analysis of risk factors for work-related musculoskeletal disorders during repetitive lifting task in construction workers. Automation in Construction, 2017. 83: p. 41-47.
[8] Bosch, T., et al., The effect of work pace on workload, motor variability and fatigue during simulated light assembly work. Ergonomics, 2011. 54(2): p. 154-168.
[9] Cardozo, A.C., M. Gonçalves, and P. Dolan, Back extensor muscle fatigue at submaximal workloads assessed using frequency banding of the electromyographic signal. Clinical biomechanics, 2011. 26(10): p. 971-976
Lines 56-58: I do not understand this sentence. Is it possible to re-write it to make it easier to understand?
Reply:L57-59-- This sentence “…compared to the task rotation, the effect of postural variation on the conventional manifestation of shoulder muscle fatigue is more significant during the experiment” has been revised as follows:
“…during the sMEG detection, the impact of postural variation, compared to the task rotation, on the conventional manifestation of shoulder muscle fatigue is obviously easier to be detected”.
Lines 115-116: The placement of the electrodes on the biceps brachii remains to be explained.
Reply:L123-126, P3-4-- We added “At 1/3 from the fossa cubit on the line between the medial acromion and the fossa cubit for the biceps brachii” and further revised all the other descriptions (all according to the related introduction from http://www.seniam.org/ and the following articles:
[36] Ghofrani, M., et al., Reliability of SEMG measurements for trunk muscles during lifting variable loads in healthy subjects. Journal of bodywork and movement therapies, 2017. 21(3): p. 711-718.
[37] Mork, P.J. and R.H. Westgaard, Long-term electromyographic activity in upper trapezius and low back muscles of women with moderate physical activity. Journal of Applied Physiology, 2005. 99(2): p. 570-578.).
Line 126: “By” in capital letters.
Reply: L135-- “By” has been corrected to “by”.
Figure 7 and 8. Do these graphs refer to the combined data of all study subjects? Actually this is not explained neither in the text nor in the figure caption / s.
Reply: L201-202, 222-223, 226-227, 244-245, 247-248, 268-269-- Actually, these figures, fig. 7-9, refer to the combined data of all subjects. The titles of these figures, and corresponding text at the first paragraph of each section, have been revised so that they can be explained clearly.
In addition, it would be very visual (to facilitate understanding of the work), to indicate on the graph (eg with asterisks) the statistically significant differences between heights (in the same muscle) and between muscles (for the same height).
Reply: We followed your suggestion and revised the formal graphs, inside each one included 3 targeted muscles data in all tasks, into a combined version as below. The revised version satisfies the visualization of statistically significant differences between heights (in the same muscle) and between muscles (for the same height).
Lines 262-264: Indicate at the end of line 263 the number of subjects (2 subjects) and at the end of line 264 (10 subjects).
Reply: L270-279, P9-10-- Sorry for that we couldn’t find the exact place you’ve pointed out. But we still checked the whole section and revised several places that can bring into misunderstanding.
Line 299-300. Please check this sentence (syntactic construction). It's not easy to understand.
Reply: L48-50, P2 & L306, P10-- In fact, it’s to respond to this sentence in “Introduction”: “…for the repetitive operation with high-frequency and low-load (for each time) during MMH tasks, the large vertical-distanced lifting is likely to cause myoelectric manifestation of muscle fatigue”.
But it seems to be hard to understand and not so inevitable for the context, so that we deleted this sentence: ”the back muscles, including the erector spinae and latissimus dorsi, fatigue more quickly during the low-load and high-frequency task than during the heavy-load and low-frequency handling task”.
Fig 10. Unnecessary. Furthermore, it is not cited in the text. Please remove it.
Reply: We deleted it according to your suggestion, as its function is nothing more than to show what semi-squatting-bending is like.
Although I think it is correct to discuss the existence of the “bend-free board-rack design”; however, I believe that such a detailed explanation of its performance is unnecessary (i.e., please simplify lines 330-335). Also, I would suggest to remove Figures 11 and 12, as they do not seem to add anything to the text of the manuscript.
Reply: L336-L339-- We followed your suggestion and simplified the detail of the “bend-free boards rack design” as follows:
“…with the cooperation of infrared-emitting and -receiving tubes, signal processor, and various valves inside the hydraulic system, the upper shelf of the rack is able to maintains at the fixed height”.
But from my point of view, the visualization of the “bend-free board-rack design” can make the description easier to understand. So that we kept

This manuscript is a resubmission of an earlier submission. The following is a list of the peer review reports and author responses from that submission.
Round 1
Reviewer 1 Report
Comments to the Authors of manuscript number: ijerph-1051186 entitled “Effects of Vertical Lifting Distance on Upper-Body Muscle Fatigue”.
Authors have presented the electrophysiological examination (EMG) in repetitive bend-handling, when the muscles on the arm, shoulder and back frequently contract and easily fatigue. It is know that EMG allows to determine the level of damage of muscle. EMG is the measurement and analysis of the electrical activity (potential) in skeletal muscles created during muscle contraction. This technique is useful for diagnosing the health of the muscle tissue and the nerves that control them.
Fatigue is a reducing ability to work caused by effort. Fatigue occurs when prolonged and strong stimulation of an exercising muscle reaches a state when the muscle is no able longer to respond to the stimulation with the same degree of contractile activity.
The performed study is very interesting, and suits to chosen Journal, however, the manuscript should be corrected in some points.
- L23- or the EMG MPFFT. Naming and abbreviations should be uniformed through the test
- L 30 – the text cannot starts from the abbreviation, when it is present for the first time (omitting abstract).
- the lack space in many places in the text
- L 67 adaption or adaptation?
- L 77 – full abbr. should be used
- the number of permission of local ethical committee should be given
- Why these 3 different muscles were chosen? Each of them shows various strength, which depends on the physiological cross section and anatomical cross section; and plays different role in the body being responsible to on different motion
- L 86 There is information about the lack of strenuous exercises, but there is no information if these participant have the same physical activity before the test? Are they training?
- L 220 – capital letter
- L 286 – in the methods the measurement of agle of the bending was not mentioned.
Author Response
Dear Reviewer:
We would like to thank you for your time and all the constructive comments. Your suggested revisions have been carefully considered, and the manuscript was revised accordingly. We believe that incorporating the suggested changes has enhanced the quality of our manuscript.
The following is the replies to the comments.
Authors have presented the electrophysiological examination (EMG) in repetitive bend-handling, when the muscles on the arm, shoulder and back frequently contract and easily fatigue. It is known that EMG allows to determine the level of damage of muscle. EMG is the measurement and analysis of the electrical activity(potential) in skeletal muscles created during muscle contraction. This technique is useful for diagnosing the health of the muscle tissue and the nerves that control them.
Fatigue is a reducing ability to work caused by effort. Fatigue occurs when prolonged and strong stimulation of an exercising muscle reaches a state when the muscle is no able longer to respond to the stimulation with the same degree of contractile activity.
The performed study is very interesting, and suits to chosen Journal, however, the manuscript should be corrected in some points.
- L23- or the EMG MPFFT. Naming and abbreviations should be uniformed through the test
Reply: L24--The EMG MPF fatigue threshold and its abbreviation have been listed as “EMG MPF fatigue threshold (MPFFT)” in the abstract.
- L30 – the text cannot start from the abbreviation, when it is present for the first time (omitting abstract)
Reply: L31-- This part has been corrected as “Manual material handling (MMH)”.
- the lack space in many places in the text
Reply: We have checked and revised carefully.
- L67 adaption or adaptation?
Reply: L74-- It’s “adaptation”.
- L77 – full abbr. should be used
Reply: L73-- This part has been corrected as “EMG MPF fatigue threshold (MPFFT)”.
- the number of permission of local ethical committee should be given
Reply: The study was conducted in accordance with the Declaration of Helsinki, while all subjects gave their informed consent for inclusion before they participated in the experiment. But we are sorry that we cannot provide ethics review approval documents. The reasons are as follows:
1) Although our experiment involves the human body, the research in this article is actually an experiment related to ergonomics instead of clinical or biomedical research. Our experiment mainly focuses on "the fatigue of upper body muscles during artificial material handling tasks". And the risks of the experiment process are no more than the minimum risk.
2) In addition, all subjects gave their informed consent for inclusion before they participated in the study, which means they understand the content of the experiment and understand the risks involved. All subjects' upper limbs returned to normal 1-2 days after the experiment.
3) Our experiment was approved by our laboratory,but we do not have a complete ethics committee, so we couldn’t apply for approval in time.
- Why these 3 different muscles were chosen? Each of them shows various strength, which depends on the physiological cross section and anatomical cross section; and plays different role in the body being responsible to on different motion
Reply: L35-37-- 1) As mentioned in the paper, no less than 39% of manual workers in various industries, among which plenty of operations are quite similar, suffered from low back pain (LBP) and neck/shoulder pain (NSP). LBP directly relates to the waist muscles, while NSP relates to the muscles of the shoulders and upper limbs. Accordingly, we have chosen the biceps brachii, upper trapezius and multifidus, which are all directly involved in the motions during manual handling, as the target muscles. Furthermore, the varying posture and contraction torque during manual handling, which are influenced by the lifting height, have an effect on work fatigue of these three muscles.
2) Actually, the physical factors such as fiber composition, contraction type greatly affects the myoelectric manifestation. However, the applying of normalized value of myoelectric parameter during data analysis, which figures out the capacity contribution of each muscle, is able to avoid the influence of these factors up to a point.
- L86 There is information about the lack of strenuous exercises, but there is no information if these participants have the same physical activity before the test? Are they training?
Reply: L92-98-- Except the training before the experiment, all of the subjects were non-experienced on the MMH tasks. While considering experience as a potential interference factor, we have reduced its impact on the experiment by simplifying the lift-and-place process.
- L220 – capital letter
Reply: L245-- This part has been corrected as “There is …”.
- L286 – in the methods the measurement of angle of the bending was not mentioned.
Reply: L155-157-- “2.3 Task and Measurements” section has supplemented how the ergonomic angle meter was applied to measure the subject’s bending angle. This instrument has been used like a protractor, one end was parallel to the thigh femur and one end was parallel to the back while measuring the bending angle.

Reviewer 2 Report
General:
The study aims to investigate muscular fatigue during a manual material handling task. The investigators chose to evaluate different lifting heights of low-load items. Besides the poor English language and style, I do have some major concerns about the rational of the study and the study design, which is why I recommend the rejection of this paper. First, the authors did not specify the movement technique used which obviously has major impact on the distribution of muscle activity. Presupposing all subjects do not squat but bend in the hip when lifting the item, it is pretty obvious that the multifidi have the highest rate of fatigue. I do not see a special scientific question of interest here. Moreover, it is very easy to use just a higher rack for the items and solve that problem, since it is common sense that bending deeper is more exhausting over time. One other major concern is, that I cannot see any approval of an ethics committee. Plus, only apparently trained young students were recruited for the study. No real carpenter participated. In total, I do not see the benefit of this study for the scientific community.
Introduction:
First you talk a lot about the rotation of the tasks and then there is only one sentence about postural adjustments, but as I get it, your study is about postural adjustments. It is not clear why you investigate the lifting height? Why is that especially of interest?
Line 41-43 : all increase the work load? The sentence is unclear, pleas clarify. How do you define work load?
Line 43: you mean a combination of low load and high frequency?
Line 44: what do you mean by high level height lifting?
Line 43-45: please rephrase for clarity.
Line 49: tenses.. must be „compared“
Line 51: reference style?
Line 66: what is EMG and MPF? You have to explain all abbreviations when you use them for the first time
Line 73: abrrev. sEMG? Pleas explain.
Methods:
Was your study appoved by an ethics committee? I did not find such information in your manuscript.
Were the subjects instructed on how exactly to carry the boards? The fatiguabilty clearly depends on the exact movement technique. Your task could be managed either with some kind of squat, or with a forward bend. Bending clearly stresses the muscles on the back more than a squat in which the muscles of the legs also contribute to the movement. If the movement was not instructed, the results can be deteriorated by these issues.
You write that you recruited your subjects from local universities and given the figures your subjects seem pretty sportive and seem to have experience in resistance training. Why didn#T you recruit actual carpenters or workers who have to perform that task regularly. It can be expected, that the fatigability of the trained and young students is not representative for the occupational group you aim to investigate.
Your MMH task involves a body twist, but you investigate only the right body side. Why didn’t you examine the left body side as well? There could be an influence due to the rotation in the movement.
Did you do 150reps for all three heights? What was the break duration between testing of the different heights? All on one day or separate days?
Line 85: check space and decide for one consistent way
Line 94: which previous studies? Reference must be placed behind the word studies.
Fig. 1: Poor resolution. Please provide higher quality pictures.
Discussion:
The whole section needs intensive rewriting and English editing. It is often unclear what you want to say.
You should not start a discussion section with a statement from other literature.
Line 245: singular or plural?
Line 252: Which are the upper body and shoulder postures in your task? You do not provide any information on the technique used. What do you mean by dynamic contraction?
Line 260: What do you mean by variable level?
Line 273: You can not make such a general statement. This is only true for your specific task.
Author Response
Dear Reviewer:
We would like to thank you for your time and all the constructive comments. Your suggested revisions have been carefully considered, and the manuscript was revised accordingly. We believe that incorporating the suggested changes has enhanced the quality of our manuscript.
The following is the replies to the comments.
The study aims to investigate muscular fatigue during a manual material handling task. The investigators chose to evaluate different lifting heights of low-load items. Besides the poor English language and style, I do have some major concerns about the rational of the study and the study design, which is why I recommend the rejection of this paper. First, the authors did not specify the movement technique used which obviously has major impact on the distribution of muscle activity. Presupposing all subjects do not squat but bend in the hip when lifting the item, it is pretty obvious that the multifidi have the highest rate of fatigue. I do not see a special scientific question of interest here. Moreover, it is very easy to use just a higher rack for the items and solve that problem, since it is common sense that bending deeper is more exhausting over time. One other major concern is, that I cannot see any approval of an ethics committee. Plus, only apparently trained young students were recruited for the study. No real carpenter participated. In total, I do not see the benefit of this study for the scientific community.
Introduction:
-First you talk a lot about the rotation of the tasks and then there is only one sentence about postural adjustments, but as I get it, your study is about postural adjustments. It is not clear why you investigate the lifting height? Why is that especially of interest?
Reply: L44-50-- Sustained muscle contraction, excessive muscle tension and extreme working postures have been considered as physical work-related risk factors. While all physical factors above are directly related to vertical and horizontal lifting distance, the vertical lifting distance, or the lifting height, could be considered as the actual controllable variable during manufacturing. Therefore, in the experiment, the lifting height was chosen to be the single test factor to simulate the real situation among manufacturing.
Furthermore, for manual handling tasks, the direct effect of lifting height, compared with the working posture, or the rotation of tasks, on muscle fatigue is more likely to figure out through experiment design.
-Line 41-43: all increase the work load? The sentence is unclear, pleas clarify. How do you define work load?
Reply: L50-53-- 1) For the mentioned situations, include sustained muscle contraction, excessive muscle tension, and maintaining extreme working postures, during which the traction, as well as the energy supply of the local muscles, will be more severely weakened than usual. In view of this, these are all considered as the situations when local muscles probably suffer from abnormal external load.
2) As an important factor of MMH task and a major cause of muscle fatigue, work load is indicated as the real-time tension of the local muscle to sustain the external load in this paper and several previous studies (Luca, 1994, Duca & Forrest, 1973, Madinei & Ning, 2017).
-Line 43: you mean a combination of low load and high frequency?
Reply: L47-50-- Yes, and the inaccurate description has corrected to “the repetitive operation with high-frequency and low-load (for each time)”.
-Line 44: what do you mean by high level height lifting?
Reply: L48-49-- it means the large vertical-distanced lifting to which we have corrected.
-Line 43-45: please rephrase for clarity.
Reply: L47-50-- We have corrected the quoted phrases based on your comments.
-Line 49: tenses... must be “compared”.
Reply: L55-- the tense has been corrected to the passive one.
-Line 51: reference style?
Reply: L58-- it has been unified with other places.
-Line 66: what is EMG and MPF? You have to explain all abbreviations when you use them for the first time.
Reply: L73-- EMG is short for “electromyography”, MPF is short for “mean power frequency”. Meanwhile we have added the full name of MPFFT, electromyographic-mean-power-frequency fatigue threshold test, at the corresponding place.
-Line 73: abrrev. sEMG? Please explain.
Reply: L80-- sEMG is short for “surface electromyography”.
Methods:
-Was your study approved by an ethics committee? I did not find such information in your manuscript. Were the subjects instructed on how exactly to carry the boards?
Reply: All subjects gave their informed consent for inclusion before they participated in the experiment. The study was conducted in accordance with the Declaration of Helsinki. But we are sorry that we cannot provide ethics review approval documents. The reasons are as follows:
1) Although our experiment involves the human body, the research in this article is actually an experiment related to ergonomics instead of clinical or biomedical research. Our experiment mainly focuses on "the fatigue of upper body muscles during artificial material handling tasks". And the risks of the experiment process are no more than the minimum risk.
2) In addition, all subjects gave their informed consent for inclusion before they participated in the study, which means they understand the content of the experiment and understand the risks involved. All subjects' upper limbs returned to normal 1-2 days after the experiment.
3) Our experiment was approved by our laboratory,but we do not have a complete ethics committee, so we were not able to apply for approval in time.
-The fatiguability clearly depends on the exact movement technique. Your task could be managed either with some kind of squat, or with a forward bend. Bending clearly stresses the muscles on the back more than a squat in which the muscles of the legs also contribute to the movement. If the movement was not instructed, the results can be deteriorated by these issues.
Reply: L39--We did realize that the variation in movement/posture, reducing bending, or replacing it with squatting, can bring totally different results. Meanwhile, instructing operators about the handling movement/posture in advance can ensure that the operation to be more ergonomic. However, in Chinese mainland, the practical problem is that most workshop foremen have no ergonomics learning experience, nor are they aware of the actual benefits of adjusting operating movements. Besides, generally the manual workers don’t have a good education, they usually unwilling to follow the specific instruction to adjust the habitual operating gestures. What is worse, work stations like this type, the turnover rate is always high.
Considering the above issues, we have restored the on-site manual handling tasks, as well as the operators’ handling movements, in the research. By this way, the fatiguability of the targeted muscles is analyzed and compared. For the most fatigable muscle, the fatigue threshold is calculated. Based on the above results, we try to make simple and effective adjustments to the operators’ habitual movements. Meanwhile, we expect to obtain a more ergonomic task rotation scheme, through JASA (joint analysis of EMG spectrum and amplitude) of the targeted muscles, in another paper. We believe that such research results and applications are more practicable to most labor-intensive factories in China than specific manual handling instructions.
-You write that you recruited your subjects from local universities and given the figures your subjects seem pretty sportive and seem to have experience in resistance training. Why didn’t you recruit actual carpenters or workers who have to perform that task regularly. It can be expected, that the fatigability of the trained and young students is not representative for the occupational group you aim to investigate.
Reply: L94-96-- 1) The subjects chosen are similar to the workers in terms of anthropometric characteristics, yet individual differences, such as muscle strength, are inevitable. And only seldom subjects are sportive. Both the operators and the students are not been trained, and we have to ignore the influence from operating experience since all the manual handling tasks are not so skillful.
2) It’s very difficult to recruit so many actual workers to be our subjects due to our insufficient funds and their insufficient free time.
-Your MMH task involves a body twist, but you investigate only the right body side. Why didn’t you examine the left body side as well? There could be an influence due to the rotation in the movement.
Reply: L161-162-- In fact, the muscles on both sides were examined and analyzed in our work, while only the analysis results of the right-sided muscles are shown in this paper. The reason for this is that, by comparison, the myoelectric manifestation of the right-sided muscles was found to be more active and variable, which was more suitable for obtaining useful conclusion. In addition, all 12 subjects are right-handed, which gives some explanation for the myoelectric manifestation of the right-sided muscles, as well.
-Did you do 150reps for all three heights? What was the break duration between testing of the different heights? All on one day or separate days?
Reply: L126-130-- 1) This pick-and-place process was continuously repeated 150 times in one cycle, for all three heights.
2) The break between two tested tasks for each subject lasted 2 days or even more. By the MVC tests and the comparison with the very first time, we were able to ensure that the subjects’ tested muscles had been completely rested.
-Line 85: check space and decide for one consistent way.
Reply: L91-92-- We have checked and corrected.
-Line 94: which previous studies? Reference must be placed behind the word studies.
Reply: L104-105-- The precious studies mean the references [26-28], and we have corrected the citation form.
-Fig. 1: Poor resolution. Please provide higher quality pictures.
Reply: L99-100-- We have changed the original figure into a higher quality one.
Discussion:
-The whole section needs intensive rewriting and English editing. It is often unclear what you want to say.
Reply: L275-349-- We have tried our best to repolish the English writing.
-You should not start a discussion section with a statement from other literature.
Reply: L275-29-- We have repolished the context by your suggestion.
-Line 245: singular or plural?
Reply: L280-281-- We have corrected that, “biceps brachii” should be singular.
-Line 252: Which are the upper body and shoulder postures in your task? You do not provide any information on the technique used. What do you mean by dynamic contraction?
Reply: L282-284-- Sorry for our unclear description. The phrase “…an effect of the upper-body posture, particularly the shoulder posture” was to illustrate that how the upper-body postured during the manual handling task ensured that the biceps brachii muscle was contracting isotonically for most of the time during the handling tasks. We have repolished the writing language to express our viewpoints more clearly.
-Line 260: What do you mean by variable level?
Reply: L292-- This referred to the lifting height, which had been set as the single variable. And we have clarified in this section.
-Line 273: You cannot make such a general statement. This is only true for your specific task.
Reply: L305-306-- We have revised the scope of this conclusion to make it more rigorous.

Reviewer 3 Report
First of all, I’d like to express my thanks for giving me the opportunity to review this paper.
This is a study that addresses an interesting topic related to ergonomics and occupational health aspects.
In general, it seems of adequate methodological quality and provides a correct presentation of the results.
En líneas generales parece de una adecuada calidad metodológica y con una presentación correcta de resultados.
I will describe various aspects to be considered by the authors:
Abstract:
In the analysis of results (lines 21-24 page 1) it would be of interest to describe the main results (data) obtained in terms of muscle tension, fatigability, and capacity according to the single independent variable of the study (vertical lifting distance), since this is the main objective of the study.
Introduction:
- The introduction adequately reflects the current state of the subject, although in my opinion it is somewhat disorderly in its presentation. It would be relevant to define very clearly, in a more orderly way, some of the basic concepts to be used in the study (muscle tension, fatigability, and capacity). For example, the concept "capacity" appears for the first time in line 53 and yet its operational definition is described in lines 65-66 (page 2).
From there, and for each one, the type of myoelectric manifestations (parameters) to be used for their measurement should be defined and referenced.
- Lines 48-51 page 2: I don't understand the phrase “rotation between MMH tasks, varying in load level, could reduce / increase fatigue compare to performing only a heavier / lighter load task”. Please explain it more appropriately.
- In this study, objective 3 “Exploring methods to relieve the muscle fatigue during handling tasks based on its development” is not possible to meet, because the methodological design and results obtained do not allow for it.
Methodology:
- Study design, subject recruitment system, and type of sampling must be defined.
- All the variables to be collected in the study must be specified. For instance, in 2 Preparation of Measurements subsection, mention is made of the “dominant hand side”; however, there is no section of the manuscript that describes all the variables collected in the study: independent, dependent, or control (for example, the dominance).
- Please check whether reference 30 (line 102 page 3) is adequate to support this part of the text.
- Vertebral levels at which the electrodes were placed for the measurement of the electromyographic activity (sEMG) of the multifidus muscles must be specified.
- Figure 3: some figures (800) and (300) appear that are not explained in the text of the manuscript.
- All aspects related to Ethical Considerations are missing, including at least: approval by the corresponding Ethics Committee (with registration number), and mention of ethical guidelines follow up (e.g. Declaration of Helsinki). Concerning Ethical Considerations, only obtaining the “informed consent” of the participants is mentioned (lines 88-89 page 2).
- It would seem more appropriate to include the information from the first two lines (lines 136-137 page 5) of section 4 Statistical analysis in section 2.2 Preparation of Measurements.
- Figure 5. The 2nd sentence included concerning “The MPF data (10 sections) are plotted as…” is part of the explanation of the general statistical analysis and not of the linear regression of the MPF of multifidus muscle in task V250 (subject 1). Therefore, in my opinion, it should be included in the text of the manuscript.
- Figure 6. Describe which muscle in the “EMG MPFFT from data of subject 8” is being measured.
- Figure 6. Describe the muscle whose “EMG MPFFT from data of subject 8” is being measured.
- Please correct typographical errors (e.g. Figure 2 last line “ankles”; line 117 page 4 “(mm2)”; Figure 3 line 3 “atthree”; Figure 5 “VD250”).
Results:
- Please check the mistakes in the description of the pairwise comparison of the MPF0 values between the different muscles (page 8 lines 206 to 209).
- Please indicate the significance values (p-values) of the statistically significant pairwise comparisons of the MPFs values.
- Please check whether the content of the phrase “Moreover, for the biceps brachii and the multifidus, only in task VD0 represents no significant difference between their muscle fatigue rates” is correct (page 9 lines 229-230).
- Table 1. Please include that this refers to the “multifidus” muscles.
Discussion:
- Please include bibliographic references in relation to the statements in lines 278-280 and also lines 280-281 on page 11.
- From the phrase included in lines 286-287 on page 11, it can be extracted that the study also measured the bending angle.
As already mentioned in comment 2 of the Methodology section of the review, please describe all the variables collected in the study and the possible analyses, where applicable, that were carried out on them (in the suitable sections).
- Page 11 (lines 193-295). In the context of this study I am not able to understand this phrase (semi-squatting lifting ???, aplying of work chair ???). Please clarify.
Conclusion:
- Linea 330 page 12. Se introduce el concepto “flexibility”, no mencionado hasta ahora a lo largo del manuscrito. Se ruega explicar su aparición en este momento o su desarrollo de forma adecuada en el apartado correspondiente o bien su eliminación, si procede.
- Line 330 page 12. The concept "flexibility" is introduced, but is not mentioned until this point throughout the manuscript. Please explain its appearance at this time, or develop it appropriately in the appropriate section, or delete it, if appropriate.
- Line 333-335 page 12: From the results described in this manuscript, it is not possible to draw this conclusion concerning the semi-squatting posture
Some of the bibliographic references cited throughout the manuscript are very old, especially in the Introduction section. Please use the most up-to-date references.
Please also review all bibliographic references in order to adapt them to the presentation format required by the journal.
There is a bibliographic reference included within the text, and not in the bibliography section (Luger et al. 2015): rectification is requested.
Author Response
Dear Reviewer:
We would like to thank you for your time and all the constructive comments. Your suggested revisions have been carefully considered, and the manuscript was revised accordingly. We believe that incorporating the suggested changes has enhanced the quality of our manuscript.
The following is the replies to the comments.
First of all, I’d like to express my thanks for giving me the opportunity to review this paper.
This is a study that addresses an interesting topic related to ergonomics and occupational health aspects. In general, it seems of adequate methodological quality and provides a correct presentation of the results.
I will describe various aspects to be considered by the authors:
Abstract:
-In the analysis of results (lines 21-24 page 1) it would be of interest to describe the main results (data) obtained in terms of muscle tension, fatigability, and capacity according to the single independent variable of the study (vertical lifting distance), since this is the main objective of the study.
REPLY: L22-25, P1--The description about the main results has been updated according to your suggestions.
Introduction:
-The introduction adequately reflects the current state of the subject, although in my opinion it is somewhat disorderly in its presentation. It would be relevant to define very clearly, in a more orderly way, some of the basic concepts to be used in the study (muscle tension, fatigability, and capacity). For example, the concept "capacity" appears for the first time in line 53 and yet its operational definition is described in lines 65-66 (page 2). From there, and for each one, the type of myoelectric manifestations (parameters) to be used for their measurement should be defined and referenced.
REPLY: L44-56, P2--We have repolished this section, the definitions, the writing logic, according to your suggestions.
-Lines 48-51 page 2: I don't understand the phrase “rotation between MMH tasks, varying in load level, could reduce / increase fatigue compare to performing only a heavier / lighter load task”. Please explain it more appropriately.
REPLY: L53-56, P2-- We have corrected this phrase to “…rotation of operating and resting in the process of the short-term MMH tasks, which frequency varying corresponds to the work load, could reduce/increase fatigue compared to performing a coherent heavier-load-levelled /lighter-load-levelled task without interruption”. In detail, in the process of the heavier-load-levelled task, the corresponding work-and-rest-rotation would reduce the operator’s muscle fatigue. On the contrary, the work-and-rest-rotation is likely to increase the operator’s muscle fatigue in the process of the lighter-load-levelled task, while complete the task coherently is a better way.
-In this study, objective 3 “Exploring methods to relieve the muscle fatigue during handling tasks based on its development” is not possible to meet, because the methodological design and results obtained do not allow for it.
REPLY: L332-349, P12-- Your consideration is reasonable. However, we have complemented some methods and suggestions, includes semi-squatting lifting way, the application of work chair, and our utility model patent: the bending-free boards rack.
Methodology:
-Study design, subject recruitment system, and type of sampling must be defined.
REPLY: L90-94, P1-P2-- The subjects, without musculoskeletal disorders or injuries in the preceding six months and strenuous exercise within 24 hours before the experiment, were random sampled from local universities.
-All the variables to be collected in the study must be specified. For instance, in 2 Preparation of Measurements subsection, mention is made of the “dominant hand side”; however, there is no section of the manuscript that describes all the variables collected in the study: independent, dependent, or control (for example, the dominance).
REPLY: L107-108, P3-- In fact, we have clarified in “2.4 Statistical Analysis” that all subjects are right-handed people. We didn’t realize that factors such as dominant hand side could be a control variable. Thus, we have listed several variables which are likely to have an impact to our experiment.
-Please check whether reference 30 (line 102 page 3) is adequate to support this part of the text.
REPLY: L470-471, P15-- We rechecked this paper, and found it was not able to support our experimental operation. And, since the viewpoints in this article “Ghofrani M, Olyaei G, Talebian S, et al. Reliability of SEMG measurements for trunk muscles during lifting variable loads in healthy subjects[J]. Journal of bodywork and movement therapies, 2017, 21(3): 711-718” have been cited in this paper, we have updated the references.
-Vertebral levels at which the electrodes were placed for the measurement of the electromyographic activity (sEMG) of the multifidus muscles must be specified.
REPLY: L115-119, P3-- We have complemented the detailed location of each electrode, as well as the corresponding reference electrode.
-Figure 3: some figures (800) and (300) appear that are not explained in the text of the manuscript.
REPLY: L149-150, L153-154, P4-- The distance between the workbench and the board rack, from one’s center point to the other’s, was 800mm. Otherwise, the subject and the assistant stand face to face with a distance of 600mm, which means the vertical distance from one’s chest to the center of the workbench is 300mm. All the above we have added to the corresponding position of the manuscript.
-All aspects related to Ethical Considerations are missing, including at least: approval by the corresponding Ethics Committee (with registration number), and mention of ethical guidelines follow up (e.g. Declaration of Helsinki). Concerning Ethical Considerations, only obtaining the “informed consent” of the participants is mentioned (lines 88-89 page 2).
REPLY: L96-98-- All subjects gave their informed consent for inclusion before they participated in the experiment. The study was conducted in accordance with the Declaration of Helsinki. But we are sorry that we cannot provide ethics review approval documents. The reasons are as follows:
1) Although our experiment involves the human body, the research in this article is actually an experiment related to ergonomics instead of clinical or biomedical research. Our experiment mainly focuses on "the fatigue of upper body muscles during artificial material handling tasks". And the risks of the experiment process are no more than the minimum risk.
2) In addition, all subjects gave their informed consent for inclusion before they participated in the study, which means they understand the content of the experiment and understand the risks involved. All subjects' upper limbs returned to normal 1-2 days after the experiment.
3) Our experiment was approved by our laboratory,but we do not have a complete ethics committee, so we were not able to apply for approval in time.
-It would seem more appropriate to include the information from the first two lines (lines 136-137 page 5) of section 4 Statistical analysis in section 2.2 Preparation of Measurements.
REPLY: We quite agree your proposal, and the information you mentioned have been supplemented at the beginning of section 2.2.
-Figure 5. The 2nd sentence included concerning “The MPF data (10 sections) are plotted as…” is part of the explanation of the general statistical analysis and not of the linear regression of the MPF of multifidus muscle in task V250 (subject 1). Therefore, in my opinion, it should be included in the text of the manuscript.
REPLY: We have complemented the corresponding description of the linear regression of the MPF.
-Figure 6. Describe which muscle in the “EMG MPFFT from data of subject 8” is being measured. Describe the muscle whose “EMG MPFFT from data of subject 8” is being measured.
REPLY: L189, P6-- The multifidus muscle of subject 8. We have supplemented the description in the corresponding figure illustration.
-Please correct typographical errors (e.g. Figure 2 last line “ankles”; line 117 page 4 “(mm2)”; Figure 3 line 3 “atthree”; Figure 5“VD250”).
REPLY: L135, P4; L137, P4; L155, P5; L185, P6-- Thanks for your careful revising, all these errors have been corrected.
Results:
-Please check the mistakes in the description of the pairwise comparison of the MPF0 values between the different muscles (page 8 lines 206 to 209).
REPLY: L231-234, P8-- We checked out the huge mistakes and have corrected them. It’s the upper trapezius has a significantly higher initial MPF value than the biceps brachii and the upper trapezius.
-Please indicate the significance values (p-values) of the statistically significant pairwise comparisons of the MPFs values.
REPLY: L252-257, P9-- We have complemented all the significance values (p-values) of the statistically significant pairwise comparisons of the MPFs values in the corresponding places.
-Please check whether the content of the phrase “Moreover, for the biceps brachii and the multifidus, only in task VD0 represents no significant difference between their muscle fatigue rates” is correct (page 9 lines 229-230).
REPLY: L253-254, L256-258, P9-- It’s correct, since the significance values p=0.35 for the difference between the biceps brachii and the multifidus in task VD0.
-Table 1. Please include that this refers to the “multifidus” muscles.
REPLY: L270-271, P10-- It’s been renamed as “The EMG MPFFT and the original muscle tension of the multifidus muscle of each subject in each tested task”.
Discussion:
-Please include bibliographic references in relation to the statements in lines 278-280 and also lines 280-281 on page 11.
REPLY: L310-312, P11; L505-509, P16-- The opinions in these two articles are quoted in this article:
Hsu C F, Lin T T. Development of an Ergonomic Evaluation System Based on Inertial Measurement Unit and Its Application for Exoskeleton Load Reduction[C]//2019 ASABE Annual International Meeting. American Society of Agricultural and Biological Engineers, 2019: 1.
Zhang C, Tang W, Yu N. Opimization of Manual Handling Operation During Furniture Manufacturing[J]. Pro Ligno, 2019, 15(4): 152-156.
-From the phrase included in lines 286-287 on page 11, it can be extracted that the study also measured the bending angle.
REPLY: L155-157-- “2.3 Task and Measurements” section has supplemented how the ergonomic angle meter was applied to measure the subject’s bending angle. This instrument has been used like a protractor, one end was parallel to the thigh femur and one end was parallel to the back while measuring the bending angle.
-As already mentioned in comment 2 of the Methodology section of the review, please describe all the variables collected in the study and the possible analyses, where applicable, that were carried out on them (in the suitable sections).
REPLY: L107-L108, P3-- We have supplemented the description of all the variables.
-Page 11 (lines 193-295). In the context of this study I am not able to understand this phrase (semi-squatting lifting ???, applying of work chair ???). Please clarify.
REPLY: L310-312, P11-- Semi-squatting lifting (as shown in the figures below): lifting the load in a semi-squatting posture, with the back straight, is able to protect the muscles on the back and waist.
L312-315, P11-- The work chair (as shown in the figure below): the chair applies to provide support for the operator’s buttock, allowing the operator to maintain a working posture between standing and leaning.
Conclusion:
-Line 330 page 12. The concept "flexibility" is introduced, but is not mentioned until this point throughout the manuscript. Please explain its appearance at this time, or develop it appropriately in the appropriate section, or delete it, if appropriate.
REPLY: L382, P13-- We determined to describe that the biceps brachii muscle was the most flexible in the process of the experimental tasks. But it seems not so necessary and not scientific, we delete it at last.
-Line 333-335 page 12: From the results described in this manuscript, it is not possible to draw this conclusion concerning the semi-squatting posture
REPLY: L505-506, P16-- But we think that combining the viewpoints of the above papers (Hsu C F, Lin T T. Development of an Ergonomic Evaluation System Based on Inertial Measurement Unit and Its Application for Exoskeleton Load Reduction[C]//2019 ASABE Annual International Meeting. American Society of Agricultural and Biological Engineers, 2019: 1.), we can get this conclusion.
-Some of the bibliographic references cited throughout the manuscript are very old, especially in the Introduction section. Please use the most up-to-date references.
REPLY: L470-471, P15; L505-509, P16-- We have tried our best to update our references as you suggesting.
-Please also review all bibliographic references in order to adapt them to the presentation format required by the journal.
REPLY: We have tried our best to adapt the citation format.
-There is a bibliographic reference included within the text, and not in the bibliography section (Luger et al. 2015): rectification is requested.
REPLY: L58, P2-- We have checked and corrected this bibliographic reference and the corresponding citation.
